# Identification of Arctic Food Fish Species for Anthropogenic Contaminant Testing Using Geography and Genetics

**DOI:** 10.3390/foods9121824

**Published:** 2020-12-08

**Authors:** Virginia K. Walker, Pranab Das, Peiwen Li, Stephen C. Lougheed, Kristy Moniz, Stephan Schott, James Qitsualik, Iris Koch

**Affiliations:** 1Department of Biology, Queen’s University, Kingston, ON K7L 3N6, Canada; pranabdasbd@gmail.com (P.D.); peiwen.li@queensu.ca (P.L.); lough@queensu.ca (S.C.L.); monizk@queensu.ca (K.M.); koch-i@rmc.ca (I.K.); 2School of Environmental Studies, Queen’s University, Kingston, ON K7L 3N6, Canada; 3School of Public Policy and Administration, Carleton University, Ottawa, ON K1S 5B6, Canada; stephanschott@cunet.carleton.ca; 4Gjoa Haven Hunters and Trappers Association, Gjoa Haven, NU X0B 1J0, Canada; gjoa@krwb.ca; 5Royal Military College, Kingston, ON K7K 7B4, Canada

**Keywords:** Arctic char, lake trout, lake whitefish, mercury, arsenic, PCBs, genomic analysis

## Abstract

The identification of food fish bearing anthropogenic contaminants is one of many priorities for Indigenous peoples living in the Arctic. Mercury (Hg), arsenic (As), and persistent organic pollutants including polychlorinated biphenyls (PCBs) are of concern, and these are reported, in some cases for the first time, for fish sampled in and around King William Island, located in Nunavut, Canada. More than 500 salmonids, comprising Arctic char, lake trout, lake whitefish, and ciscoes, were assayed for contaminants. The studied species are anadromous, migrating to the ocean to feed in the summers and returning to freshwater before sea ice formation in the autumn. Assessments of muscle Hg levels in salmonids from fishing sites on King William Island showed generally higher levels than from mainland sites, with mean concentrations generally below guidelines, except for lake trout. In contrast, mainland fish showed higher means for As, including non-toxic arsenobetaine, than island fish. Lake trout were highest in As and PCB levels, with salmonid PCB congener analysis showing signatures consistent with the legacy of cold-war distant early warning stations. After DNA-profiling, only 4–32 Arctic char single nucleotide polymorphisms were needed for successful population assignment. These results support our objective to demonstrate that genomic tools could facilitate efficient and cost-effective cluster assignment for contaminant analysis during ocean residency. We further suggest that routine pollutant testing during the current period of dramatic climate change would be helpful to safeguard the wellbeing of Inuit who depend on these fish as a staple input to their diet. Moreover, this strategy should be applicable elsewhere.

## 1. Introduction

The accumulation of anthropogenic contaminants in food fish has a long and tragic history as well as being a current global health concern. International conventions and risk management policies have been introduced to address individual contaminants, including those derived from burning fossil fuels or industrial infrastructure or processes that involve electrical transformers and plasticizers in the case of mercury (Hg) and persistent organic pollutants, respectively [1,2,3]. However, such regulations are typically not comprehensively enforced and can lag decades behind problems arising from industrial innovations, as reflected in present concerns over nanomaterial and microplastic pollution [4,5]. Even when governments embrace risk-management principles and institute new regulations, contaminants may persist in waterbodies for decades, centuries, and even millennia. Of pressing concern is the identification of anthropogenic pollutants that could impact the traditional lifestyles of Indigenous peoples who depend on locally sourced food fish. Here, innovations in digital labeling, mapping Indigenous Traditional Knowledge (ITK), as well as geographical and genomics approaches have been explored in the effort to identify fish at risk for contaminants near an Arctic Inuit community.

The Arctic is frequently viewed by those in the lower northern latitudes as a region that is not only remote but is relatively unsullied by industrial activity. This is not true. For example, Hg moves in the atmosphere from emission sources in industrialized and coal-burning nations and is oxidized to a reactive Hg^2^ form during the polar sunrise, after which it accumulates on ice and snow, and from there it moves to waterways where it is transformed by microbes to neurotoxic methylmercury [6,7,8,9,10]. Similarly, anthropogenic activities including coal-fired generators, mining, pesticide usage, and industrial waste contribute to arsenic (As) contamination, with 40% of the yearly flux to the ocean deriving from the atmosphere [11,12]. The transport of persistent organic pollutants, including neurotoxic, carcinogenic, and hormone-disrupting polychlorinated biphenyls (PCBs) from southern to northern latitudes can occur via long-range atmospheric, oceanic, and riverine processes [3,13,14,15]. PCBs were initially manufactured in the 1920s and used in dielectric and coolant fluids in transformers, capacitors, and electric motors as well as plasticizers and include more than 200 compounds or congeners that differ in the number and the position of chlorine atoms on a biphenyl molecule [16]. They were targeted for worldwide elimination by the Stockholm Convention on Persistent Organic Pollutants 20 years ago but remain in virtually every part of the Arctic ecosystem and food webs, derived from long-range transport as well as local sources including those associated with cold-war military installations [17,18,19]. PCBs, As, Hg, as well as other contaminants bioaccumulate and biomagnify in fish [19,20,21].

The level and the distribution of these and other anthropogenic pollutants in Arctic fish are hard to predict. This is because concentrations are influenced by geographical and biological variables, with additional concerns of increasing pollutant burden due to climate change-mediated thawing permafrost and the alteration of drainage basin hydrology [22,23,24,25,26]. Despite these many challenges, it is important to identify those Arctic food fish species that may bear higher contaminant loads. Modest funding has been made available to assay certain species, but the costs of remote fishing expeditions, sampling, and sophisticated chemical analysis for even a few contaminants are often prohibitive. For example, until recently, in Gjoa Haven, an Inuit hamlet on the Lower Northwest Passage, only “a few individuals” of Arctic char and no other food fish species were tested for Hg, and that was 16 years ago (summarized in [27]). Clearly, other approaches are warranted. We now posit that geographical and genomics analyses may allow for fish important for local subsistence harvest to be identified and thus selectively targeted for contaminant analysis so as to utilize limited resources more effectively.

As indicated, Arctic geography may contribute to contaminant exposure and accumulation, but tools to utilize ITK understanding of the land as well as sample tracking protocols may be unreliable, and thus the range and the movements of some migratory species could be largely unknown. However, DNA markers (i.e., microsatellite and mitochondrial sequences) or genome-wide single nucleotide polymorphisms (SNPs) can delineate distinct populations or stocks (e.g., [28,29]). Encouragingly, genetic population structure has been shown in Arctic salmonids with DNA markers [30,31,32]. With advances in sequencing techniques, genome-wide SNP panels have become widely-used as markers to quantify population structure as well as identify natal origins of individuals in a mixed-stock to known populations, as can occur when fish are netted in marine environments [33,34]. We suggest that either population or, alternatively, river and lake “home water” assignment analysis using SNP panels, could be informative for fish populations and thus useful for contaminant analysis.

Here, we investigated Hg, As, and PCB contamination in several important subsistence salmonids that were caught at fishing sites on or near King William Island, in the Kitikmeot region of Nunavut, Canada. According to ITK and as well as otolith-determined age-growth plots [35] (with other data not shown), the sampled fish are anadromous, migrating up rivers prior to sea ice formation in the autumn and down again in the spring to feed in coastal waters. The focal species, *Salvelinus alpinus* (Arctic char), *S. namaycush* (lake trout), *Coregonus clupeaformis* (lake whitefish), and the ciscoes, including *C. autumnalis* (Arctic cisco) and *C. sardinella* (least cisco), were chosen because they are targeted for local consumption both on the island and adjacent mainland sites, although the ciscoes are principally by-catch. It was our hypothesis that population genomic patterns combined with a geographical analysis would correlate with levels of anthropogenic contaminants and thus be a surrogate for impractical and expensive chemical analysis of hundreds of random fish samples. It is important to note that this exploratory evaluation represents an initial assessment, designed to determine the merit of such a novel strategy, rather than a definitive prescription to safeguard the health of consumers.

## 2. Materials and Methods

### 2.1. Indigenous Knowledge, Mapping of Fishing Locations, and Sampling Methodology

Fishing sites were identified by Gjoa Haven ITK and community fishers in an area of about 67 000 km^2^ on or adjacent to King William Island and south of the Adelaide Peninsula in the Kitikmeot Region of Nunavut Canada. Indeed, the project was initiated by the Hunters and Trappers Association and supported by Elders in the community, with these groups continuing to guide the process. New approaches to the gathering of ITK facilitated the identification of fishing locations as well as traditional methods and practices [36]. Fish were sampled in December–June under ice and August–September in open water using nets or occasionally with hand lines or spears as described previously [37]. Additional “non-traditional fishing sites” were used to sample Arctic char and lake whitefish to enrich the genetic analysis. Licenses to fish for scientific purposes were obtained in accordance with Section 52 of the general fishery regulations of the Fisheries Act, Department of Fisheries and Oceans Canada (DFO). These and animal care permits were issued by the Freshwater Institute Animal Care Committee of DFO (S-18/19-1045-NU and FWI-ACC AUP-2018-63).

After capture, fish were assigned a unique numeric barcode identifier [38], photographed, measured for fork length, weighed, and otoliths were dissected and subsequently dried for age analysis [35,39]. Muscle and other tissue samples including fin clips for contaminant and genomic DNA isolation were placed in barcode-tagged sterile whirl-paks^®^, empty tubes, or tubes containing 70% ethanol, respectively, and frozen at –20 °C and shipped with freezer packs to Queen’s University for analysis, as previously detailed [32,37]. The remainders of the dissected fishes were returned to local community members after processing or distributed from the Gjoa Haven Hunters and Trapper’s community freezer.

### 2.2. Contaminant Analysis

Muscle tissue was used for all contaminant analysis. Frozen muscle samples (200 mg; ~2 cm^3^) from sampled salmonids were oven dried overnight at 70 °C or at 24 °C for two days for analysis of inorganic elements and Hg, respectively. Samples from a total of 531 fish were individually ground and submitted to the Queen’s University Analytical Services Unit (QUASU; Kingston, ON, Canada). They were analyzed for 59 elements, including As but excluding Hg, by acid digestion followed by measurements using inductively coupled plasma mass spectrometry (ICP-MS), except for boron (B), phosphorus (P), and sulfur (S), which were measured using ICP-optical emission spectroscopy (OES). Analysis of total Hg (from a total of 540 fish) was accomplished by thermal decomposition of the solid sample, amalgamation, and atomic absorption spectroscopy in an Hg analyzer (DMA-80). Economic considerations dictated that PCBs were only analyzed in 20% (101/531) of the sampled fish analyzed for inorganic elements, but took into account age, species, and fishing site in an effort to target 10 samples for each site and species, although this was not always achieved. Samples were processed (dried and ground) as for Hg analysis (but as separate subsamples). Samples (1–2 g) were analyzed at QUASU for lipids and for PCBs; the latter analysis was by extraction and gas chromatography-electron capture detection (GC-ECD) and gas chromatography-tandem mass spectrometry (GC-MS/MS) analysis for individual congeners using standard procedures (USEPA, 8082A). Quality assurance/quality control steps included analytical duplicates, certified reference materials (TORT-2) and appropriate controls and blanks for inorganic elements and Hg, and for PCBs, analytical duplicates, controls, blanks, and surrogates.

Arsenic speciation analysis was carried out by ALS Global Environmental, Vancouver, British Columbia, Canada. The dried, ground fish samples were further homogenized and then extracted using a methanol/enzymatic (protease, alpha amylase and lipase in 25% methanol solution) extraction procedure at 37 °C. The extracts were analyzed using anion exchange high performance liquid chromatography-ICP-MS. It should be noted that QUASU and ALS Global are accredited by the Canadian Association for Laboratory Accreditation Inc. to the standards of ISO/IEC 17025. Methods used were listed in the scopes of accreditation at the time of analysis.

Corrections to wet weight (ww) values when required (Hg, PCBs) were made using percent moisture obtained from a subset of samples according to fish species, either Arctic char, cisco, lake whitefish, or lake trout. All results were shared with community members in meetings with the Gjoa Haven Trappers and Hunters Association as well as the community, and additional fishing sites were recommended at each discussion, making ITK, site selection, and chemical analysis an interactive process.

### 2.3. Statistical Methods for Contaminants

Hg (as ww concentrations) and As (as dry weight; dw) concentrations, according to conventions in the contaminant literature, were analyzed for differences between means using non-parametric Kruskal–Wallis statistics, as analysis of variance (ANOVA) on log-transformed data that did not yield normally distributed residuals. Spearman rank correlations (a non-parametric measure of correlation) were calculated between age, Hg, As, selenium (Se), and nutritional elements including calcium (Ca), chromium (Cr), cobalt (Co), copper (Cu), iron (Fe), magnesium (Mg), manganese (Mn), P, potassium (K), sodium (Na), S, vanadium (V), and zinc (Zn). All statistical tests for Hg and As were performed in XLSTAT 2020.4.1, where As concentrations below the detection limits were imputed using log-normal regression on order statistics (ROS) methods in ProUCL 5.1.

For the PCB analysis, data included the PCB congeners, inorganic elements, site, collection date, species, length, age, and % lipid. Where the reported congener datasets differed (because of different resolution of congeners on the GC column), separately reported congeners were summed to obtain the same congener dataset for each sample. Calculations were determined for total PCBs, total PCBs on a lipid weight basis (T-PCBlip), dioxin-like PCBs (DL-PCBs), non-dioxin-like PCBs, and the sum of six European Union (EU) PCBs (PCB6); details are provided (Appendix A). PCB and inorganic element concentrations, including Hg, were in dw unless compared to published guidelines, as previously indicated, where ww was used. The data used for statistical analyses of PCB concentrations, performed with XLSTAT unless noted otherwise, included substitutions for non-detectable values obtained using log-normal ROS methods in ProUCL 5.1. ProUCL 5.1 was also used for two sample testing, as it allows for inclusion of non-detectable values and non-parametric approaches. Spearman rank correlations were calculated between age, % lipid, total PCBs, T-PCBlip, DL-PCBs, and inorganic elements with > 5 detectable values. Principal components analysis (PCA) was conducted with PCB congeners only (proportions, Spearman correlations) and with PCB total parameters and other parameters including inorganic elements, age, and % lipid (log-transformed, Pearson correlation). Non-parametric testing of means of PCB concentrations was conducted using the Kruskal–Wallis test on ranks, and ANOVA was conducted on log-transformed data to test for differences between geographic location groups and fish species. ANOVA results indicated that, for most of the PCB data, residuals were not normally distributed (even after log transformation), and therefore non-parametric Kruskal–Wallis testing was used.

### 2.4. Genetic and Bioinformatic Analysis

DNA was extracted from fin clips and muscle samples from 429 Arctic char samples using either a Qiagen DNeasy Blood and Tissue kit (Qiagen, Venlo, The Netherlands) following the manufacturer’s protocol or a salt extraction method [40]. DNA sample purity and concentration were assessed with a spectrometer and fluorometer as well as by electrophoresis as described [32]. Double-digest restriction associated DNA sequencing (ddRAD-Seq) libraries were constructed at the Institute for Integrative Systems Biology (Laval University, Quebec City, QC, Canada) using the restriction enzymes, *SbfI* and *MspI*. In the last step of library preparation, samples with unique barcodes (unique short sections of DNA to distinguish among individual samples) were pooled and then sized with a BluePippin^®^ DNA size selection system (Sage Science Inc., Beverly, MA, USA). SNP fragments of the appropriate size were then sequenced as single-end, 100 bp reads on a HiSeq2000 platform (Illumina, San Diego, CA, USA).

Filtering and SNP calling have been described in detail [32] with sequences submitted to the National Center for Biotechnology Information Sequence Read Archive (NCBI SRA) database under BioProject accession # PRJNA680999. Briefly, the libraries were demultiplexed and aligned to a reference genome (Arctic char GenBank accession: GCF_002910315.2; [41]), with variant calling and genotyping performed using SAMtools (v1.9) and BCFtools (v1.9; [42]) to obtain a SNP dataset, which then was filtered using VCFtools (v0.1.14; [43]). As reported [32], the final data set included 413 samples and 3055 SNPs. These were analyzed using a suite of population genetic approaches and showed population structure in the Lower Northwest Passage with genetic division between King William Island and the southerly mainland sites (and thus designated “Northern” and “Southern” populations). Here, population assignments were performed, and the power of the SNP panel to assign an individual back to its genetic cluster or, alternatively, its site of capture, was tested. In the first test, the 413 Arctic char DNA samples were tested to determine if samples taken at random could be reliably assigned to the one of the two populations (“Northern” vs. “Southern”). In the second test, the assignment of random samples to 17 individual fishing sites was undertaken. For both tests, missing data were also imputed in the SNP datasets based on allele frequencies per group using the function *RandomForestRC* [44] as implemented in R program grur [45] with 100 random trees and 10 iterations. There were thus two SNP datasets for each test, one with original data including missing values, and the other with imputed data. The program *gsi_sim* [46,47] from the R package AssigneR [48] was used to perform the assignments on both the original and the imputed datasets. Assignment tests to the two populations required 7 steps: (1) a subset of samples (N = 174) was randomly chosen from each of the two groups to eliminate any bias due to uneven sample size; (2) 50% of the samples were randomly picked to form a training dataset, and the remaining 50% comprised the holdout dataset; (3) markers were ranked based on F_ST_ values computed from the training dataset, and the top 2, 4, 8, 16, 32, 64, 128, 200, 500, 1000, 2000, and 3055 markers were used as panels of loci to evaluate the impact of the number of markers used for assignment; (4) the holdout samples were then probabilistically assigned to either “Northern” or “Southern” populations; (5) steps 2–4 were repeated 30 times; (6) steps 1–5 were repeated three times (thus for each data set, there were three replicates each with 30 iterations); and (7) the average performance was then assessed to represent assignment accuracy. Theoretical assignment to 1 of 17 fishing sites did not use steps 1 and 6 due to small sample sizes at some sites, but the rest of the steps were followed for both original and imputed datasets.

## 3. Results

### 3.1. Mapping Subsistence Fishing Locations and Sampling

ITK proved crucial for the mapping of subsistence fishing locations, including local context as to routes, equipment (hand lines, spears, and nets), as well as optimal times for western sampling protocols and resulted in robust knowledge exchange [36,49]. The mapping tool and the interactive online atlases facilitated community validation by Elders and harvesters and resulted in the sampling at more than 10 traditional subsistence fishing sites, excluding sampling done at some closely adjacent sites. Contaminant analysis was conducted on 540 salmonids (~200 Arctic char, as well as ~100 each of lake trout, lake whitefish, and ciscoes, with a few fish not yielding data for particular contaminants), and the genetic analysis used an additional ~200 more Arctic char, including those from non-traditional fishing sites, to augment the SNP analysis. Harvest information results were made available to the community and can be visualized in the Gjoa Haven Nattilik Heritage Centre using a large touch screen, thus enabling the use of this knowledge and extension with an initial Traditional Land Use and Ecological Knowledge Atlas followed by a Commercial Quotas and Opportunities Atlas. These were further annotated by community members and harvesters, where additional sites of subsistence fish were identified. Relevant aspects of these atlases are presented as a single map, on which geographical groupings of the fishing sites have been placed as has also been made available for public access (Figure 1; also https://tsfn.gcrc.carleton.ca/index.html?module=module.tsfnatlas.quotas). Additional details of each fishing site including site number, location, equipment used, water type, and if fish from a particular site were sampled for contaminants and/or genetics are shown in Table 1.

There were differences in average age, weight, and growth rates depending on the salmonid species. Arctic char sampled ranged in age from 5–29 years with a mean of 14.2 ± 0.6 years (95% confidence limits), with an average weight of 3.086 ± 247 g and a linear growth rate of 47.1 ± 1.9 mm·year^−1^. The lake trout samples representing the other *Salvelinus* species were generally older and slower growing, ranging from 8–62 years (mean = 25.4 ± 1.6), weighed 3.308 ± 186 g with a growth rate of 27.8 + 1.4 mm·year^−1^. Of the *Coregonus* salmonids sampled, the ciscoes were similar in age to Arctic char, ranging from 2–30 years (mean = 14.8 ± 1.4) but much smaller with an average weight of 504 ± 66 g and a growth rate comparable to lake trout at 28.5 ± 2.9 mm·year^−1^. The other *Coregonus*, lake whitefish ranged from 4–47 years (mean = 21.3 ± 2.2), with a weight of 935 ± 79 g and the slowest growth rate, perhaps reflecting their existence at the northern edge of their range at 23.8 ± 2.8 mm·year^−1^. All data have been deposited into the Polar Data Catalogue and shared as open access (PDC#312992; NA profile of IOS 19115:2003 uploaded 5 February 2020. doi.org/10.21963/12992). Fish samples have been archived for access.

### 3.2. Variation of Hg, As, and PCB Concentrations by Fish Species

Contaminant levels varied depending on the age of the sampled salmonids. As the fish aged, there was some accumulation of inorganic elements including Hg, independent of fish type (Figure 2). Correlations (Spearman test, degrees of freedom (DF) = 529, *p* < 0.0001 unless stated otherwise) (Appendix A) were observed between Hg and age (ρ = 0.603), As (ρ = 0.212) and other elements including Ca (ρ = 0.121, *p* = 0.005), Cr (ρ = −0.280), Cu (ρ = −0.311), Mg (ρ = 0.282), Mn (ρ = 0.184), P (ρ = 0.266), K (ρ = 0.301), Na (ρ = 0.478), S (ρ = 0.518), V (ρ = −0.261), and Zn (ρ = 0.220). Fewer correlations (*p* < 0.0001 unless indicated) were observed for As, specifically with age (ρ = 0.403), Ca (ρ = −0.116, *p* = 0.007), Cr (ρ = −0.273), Cu (ρ = −0.102, *p* = 0.019), P (ρ = −0.085, *p* = 0.049), Na (ρ = 0.273), and V (ρ = −0.142, *p* = 0.001). As levels correlated with Se (ρ = 0.265, *p* < 0.0001), but Hg did not (ρ = −0.031, *p* = 0.478). Similarly, as a group, older fish showed increased PCB levels (Figure 2). Total PCBs correlated (Spearman test, DF = 99) (Appendix A) positively with age (ρ = 0.263, *p* = 0.008), Se (ρ = 0.258, *p* = 0.009), Hg (ρ = 0.234, *p* = 0.019), and Na (ρ = 0.296, *p* = 0.003) and negatively with Cr (ρ = −0.265, *p* = 0.008) and V (ρ = 0.198, *p* = 0.048), but no correlations were seen with % lipid (ρ = −0.035, *p* = 0.727). When PCBs were expressed on a % lipid basis (T-PCBlip), positive correlations were again seen with age (ρ = 0.401, *p* < 0.0001), Hg (ρ = 0.438, *p* < 0.0001), and Se (ρ = 0.285, *p* = 0.004), as well as with As (ρ = 0.218, *p* = 0.029). Correlations of T-PCBlip (*p* < 0.0001 unless indicated) were observed with the nutritional elements Zn (ρ = 0.234, *p* = 0.002), S (ρ = 0.535), Na (ρ = 0.545), Mg (ρ = 0.281, *p* = 0.005), P (ρ = 0.402), K (ρ = 0.397), and Mn (ρ = 0.222, *p* = 0.026) as a consequence of the strong negative correlation of these nutritional elements with % lipid (ρ = −0.397 to −0.778, all *p* < 0.0001). That is, fewer fatty fish, which result in higher T-PCBlip values, have higher concentrations of these nutritional elements. Within the PCB and associated inorganic elements dataset, Hg negatively correlated with % lipid (ρ = −0.497, *p* < 0.0001), but As did not (ρ = −0.168, *p* = −0.93).

Not all the salmonids accumulated contaminants to the same levels. The mean Hg in Arctic char in the sampled waters was 0.07 mg∙kg^−1^ ww, with similar concentrations for lake whitefish (0.11 mg∙kg^−1^ ww) and ciscoes (0.09 mg∙kg^−1^ ww), although the average concentrations in the older salmonids were higher (Figure 2A). Lake trout Hg levels were also statistically higher in older fish (2 sample *t* test, DF = 140, *p* < 0.0001) and across all age classes had a higher mean level of 0.36 mg∙kg^–1^ ww, statistically higher than the mean Hg concentration for all other fish (Kruskal–Wallis, *n* = 540, DF = 3, *p* < 0.0001). Differences were seen between the overall means of other fish species as well (*p* < 0.0001), except for ciscoes and lake whitefish.

Overall, As levels in the sampled fish varied over a broad range of concentrations (<0.5–270 mg·kg^−1^ dw), and a wide range was also seen when individual species were examined (Figure 2B). A positive association of fish age with As levels was seen for all the salmonids except for the ciscoes (2 sample *t* test, DF = 201, *p* < 0.0001 for char, DF = 135, *p* < 0.0001 for lake trout, and DF = 102, *p* = 0.04 for lake whitefish). As levels varied depending on the species, with the highest average values (25 mg·kg^−1^ for young fish and 59 mg·kg^−1^ for older fish) found in lake trout, with the overall mean significantly higher in these fish than in the other species (Kruskal–Wallis, *n* = 537, DF = 3, *p* < 0.0001). The lowest values were seen in lake whitefish (3.9 mg·kg^−1^ in young and 5.1 mg·kg^−1^ in older fish), with the overall mean significantly different from the other fish species (*p*< 0.0001 to *p* = 0.008); means for Arctic char and cisco were not significantly different from each other.

Overall, the mean PCB concentration was 15 µg·kg^−1^ ww. However, since only 20% of the fish processed were assayed for PCBs due to costs, comparisons of contaminant concentrations with age were somewhat limited (Figure 2C). Nevertheless, similar to the overall As results, the values varied over a wide extent (0.04–367 µg·kg^−1^ ww) with Arctic char showing the broadest absolute range of all the salmonids (0.15–367 µg·kg^−1^, means of 5.6 in young and 27 µg·kg^−1^ in older fish). Lake trout had mean values of 7.1 in young and 26 µg·kg^−1^ in older fish, and, curiously, PCB levels in lake whitefish showed a reverse of the general trend for contaminant levels in all fish species in that they had higher mean values in young fish compared to older fish (20 vs. 7 µg·kg^−1^, respectively; Figure 2C). It should be noted that, since the PCB sample sizes were necessarily small, these differences were not statistically significant.

### 3.3. Geographical Analysis of Contaminant Levels

To understand site-specific contaminant levels and to keep sample numbers sufficiently high, all of the fishing sites were divided into four groups based on their geographic locations (Figure 1). When fish were grouped irrespective of species, group 2 fish, located on the island close to Gjoa Haven, had significantly higher Hg levels than group 1 and mainland group 4 fish (Kruskal–Wallis, *n* = 540, DF = 3, *p* = 0.022, 0.027), with no significant differences between group 3 fish and any other groups or between group 1 and 4 fish. Species-specific comparisons showed that Arctic char obtained from nine fishing sites and grouped into four geographic regions showed low Hg levels independent of location, ranging from 0.05–0.13 µg·g^−1^ ww (Figure 3A). Lake trout fished from sites placed in three groupings showed the highest levels ranging from 0.14–1.36 µg·g^−1^ ww. *Coregonus* species generally showed low levels at all locales (0.02–0.13 µg·g^−1^ ww), but ciscoes from King William Island sites had significantly (*p* < 0.01) higher mean Hg levels than those obtained from mainland sites (groups 1 and 2 vs. 3 and 4) at 0.16 and 0.08 µg·g^−1^ ww, respectively. It should be noted that community members were interested in obtaining contaminant information from all individual sites, but this was not possible. For life history or logistical reasons, individual species were not obtained or were taken in insufficient numbers at all sites, particularly since this region represents the northern most distribution of lake whitefish [50].

In contrast to the generally species-specific results for Hg levels, clear differences were apparent when As levels from all salmonids obtained from mainland fishing sites were compared to King William Island sites (groups 3 and 4 vs. group 1 and 2), and these were statistically significant (Kruskal–Wallis, *n* = 537, DF = 3, *p* = 0.861 for groups 1 and 2, *p* = 0.997 for groups 3 and 4, *p* < 0.0001 between the island sites and the mainland sites). This geographic trend was also seen for Arctic char, lake trout, and cisco (Kruskal–Wallis, *n* = 203, DF = 3 for char; *n* = 137; DF = 2 for lake trout; *n* = 93, DF = 3 for cisco, and *p* ≤ 0.001), but only the group 1 lake whitefish mean, and not group 2, was significantly different from the means of lake whitefish groups 3 and 4 (*n* = 104, DF = 3, *p* ≤ 0.002). Additionally, the mean As in Arctic char from group 3 was significantly lower than that of group 4 (*n* = 203, DF = 3, *p* = 0.048), but no differences were seen between group 3 and 4 means for other fish (Figure 3B).

When PCB levels for all the fish species were combined, no statistically significant differences were seen overall between the different geographical groups. Arctic char samples analyzed at all four geographic location groups similarly showed no statistical differences for total PCBs (Figure 3C). However, DL-PCB content from Arctic char caught from sites within the island group 1 was significantly lower than mainland group 4 (Kruskal–Wallis, DF = 3, *p* = 0.03) as well as mainland groups (3 and 4) in T-PCBlip (Kruskal–Wallis, DF = 3, *p* = 0.002; Figure 3C). PCB concentrations from ciscoes in group 1 and 4 were not significantly different for total PCBs and T-PCBlip and for DL-PCBs, but lake whitefish from group 1 were significantly higher than those from group 4 (2-sample t test, DF = 21, *p* = 0.01 for total PCBs, 0.003 for T-PCBlip, and 0.01 for DL-PCBs; Mann–Whitney test gave similar results). PCB content differences between species were not apparent at location group 4, the only location group for which all fish species were analyzed. On the other hand, Arctic char PCB concentrations in group 1 were significantly lower than group 1 lake whitefish (Kruskal–Wallis, DF = 3, *p* = 0.03 for total PCBs, T-PCBlip and DL-PCBs) and ciscoes (no difference, *p* = 0.086 for total PCBs, 0.013 for T-PCBlip, and 0.029 for DL-PCBs). Only Arctic char were analyzed from location groups 2 and 3 (Figure 3C).

Exploratory PCA analysis indicated different fingerprints on the basis of PCBs, lipid, age, and inorganic elements with Arctic char samples from group 1 and 2, cisco and lake whitefish from group 1, and lake whitefish from group 4 plotting somewhat separately from each other but with a general overlap for all Arctic char and lake trout (Appendix A). The PCA was influenced by correlations between PCBs, Hg, As, age, and some nutritional elements, and, interestingly, a lack of correlation between PCBs and lipid as well as a negative correlation between age and lipid (older fish are leaner but contain higher concentrations of PCBs). PCB congener fingerprints again using PCA analysis indicated that Arctic char from island fishing sites (groups 1 and 2) plotted separately from each other and generally from the rest of the fish (Appendix A). Three lake whitefish samples from group 1 were clustered together but plotted slightly away from other lake whitefish samples (group 4), which plotted in the center of the plot with the other fish from group 4. Different locations on the PCA plot suggest differences in congener profiles, albeit with profiles in groups 1 and 2 for Arctic char and group 1 for lake whitefish showing the same predominant congeners, characterized by five or six chlorines (Appendix A). The differences were seen in Arctic char having higher levels of lower chlorinated congeners (2–4 chlorines) as well as higher chlorinated congeners (8–9 chlorines) than lake whitefish from group 1. Strikingly, the lower and the higher chlorinated congeners were almost absent in lake whitefish. PCB congeners 110, 153, 118, and 138 represented the largest peaks, and three or more of these peaks were amongst the highest five peaks in 84% of all samples (Appendix A).

### 3.4. Arctic Char Geographical and Population Assignments as Revealed by Genetics

Population assignments were undertaken for Arctic char using 3055 SNP markers, with one population that included fish caught on or near King William Island and designated as “Northern” residents and a mainland or “Southern” population ([32]. Random samples of 413 Arctic char DNA samples were successfully assigned to the one of the two populations using as few as 32 SNPs, producing > 90% correct assignments overall. However, even as few as four SNPs generated > 85% correct assignment (Figure 4A). Reassuringly, when the analysis was repeated three independent times, the results were unchanged and remained so even when missing data were imputed (Appendix A). Less than 13% of individuals sampled from the “Southern” population (21/174 in the dataset with no data imputation and 22/174 in dataset with missing data imputed) were mis-assigned as “Northern” Arctic char, while all 174 DNA samples from “Northern” fish were successfully assigned back to this population (Figure 5A; Appendix A).

In contrast, the designation of individual Arctic char to particular fishing sites was much less successful either with non-imputed or imputed datasets (Figure 4B, Figure 5B, Appendix A). Depending on the number of markers, the overall correct assignment for the fishing site-level analysis varied from 10% (two SNPs) to 50% (500 SNPs) (not shown). Three fishing sites showed reasonable assignment success (>75%) with the full SNP panel (Figure 4A). Alternatively, rather than using the full panel, if only 200 SNPs with the highest computed F_ST_ values were used, assignment to 35% of the fishing sites achieved a success rate of >75%. Several sites showed close to 100% mis-assignment no matter the number of markers used (Figure 4B and Figure 5B).

## 4. Discussion

### 4.1. Distribution of Contaminants by Species and Geography

Fish are a dietary staple for Indigenous communities living along the Arctic coasts. According to a wildlife harvest study, Gjoa Haven residents consumed 9279 Arctic char, 2427 lake trout, and 4080 whitefish and cisco annually [51]. Therefore, we calculate that more than two servings of these locally harvested species are consumed per person per week, given an estimated total edible weight of approximately 22,500 kg and an average population of 920 [52,53,54]. Thus, the identification of Arctic food fish that bear high contaminant loads is crucially important for Inuit who depend on these resources for subsistence. However, costs of these analyses need to be balanced against funding demands to address numerous twenty-first century challenges including social transformation, increases in population, climate change-mediated ice and permafrost thaw, in addition to anthropogenic pollutants. This is an important consideration since, as previously mentioned, only a single contaminant had been tested in “a few” Arctic char from the community under study 16 years ago. We considered that geographical and genomic analyses would offer the prospect of directed contaminant testing, if targeted to anadromous food fish in the Arctic Ocean, and, if successful, this strategy would be applicable to similar challenging environments elsewhere.

We selected Hg, As, and PCBs for assay at the behest of the people of Gjoa Haven, an Inuit community on the shores of the lower Northwest Passage, who, as indicated, rely on anadromous salmonids as an important part of their traditional diet. Success in sampling depended upon a respect and integration of ITK and interactive mapping and atlases as well as sample tracking using a newly developed R-based barcoding system [36,38]. A large number of samples were obtained and, overall, 6% (31/540) of the fish tested for Hg exceeded Canadian guidelines for commercial sale (0.5 mg·kg^−1^ wet weight) [55], with preliminary analysis of a few fish showing that much of the Hg was in the neurotoxic methylated form. Notably, all but one of the exceedances were from lake trout, representing ~21% of the trout tested. Since these fish are not marketed but consumed entirely within the community, a ~0.2 mg·kg^−1^ recommendation for subsistence consumption may be more appropriate, similar to those issued by several jurisdictions, including American states (e.g., Alaska advisory for women and children at 0.15 mg·kg^−1^ [56], sport fish in California at 0.2 mg·kg^−1^ [57], the US New England region at 0.2–0.3 mg·kg^−1^ [58]; and Canadian provinces (e.g., consumption limit advice for fish levels between 0.2–0.5 mg·kg^−1^ [59]). Although contaminant levels were generally higher in older fish, irrespective of species or contaminant type (Figure 2), for Hg, 70% of all lake trout sampled exceeded this level, even for the youngest fish harvested (8 years old). Few (6–8%) of the other salmonids exceeded this lower benchmark. Lake trout, unlike adult Arctic char, which fast during the winter, can be caught on baited lines under the ice and thus are a popular source of protein for the Gjoa Haven Hunters and Trapper’s “food bank” freezer as well as groups such as “moms and tots”, “senior lunches”, and “prenatal cooking classes”. The high Hg levels in lake trout and the absence of regional or territorial consumption guidelines are thus cause for concern.

Bioaccumulation of high Hg levels in lake trout undoubtedly reflect the positive correlation between contaminant levels and fish weight [60], but, in addition, they are piscivores and feed throughout the year seasonally in both fresh and marine waters, in contrast to the overwinter fasting by Arctic char with similar average weights. Community members were concerned about Hg pollution and wondered if it was regionally governed, in that fishing sites, either close to Gjoa Haven or more distant, might vary in concentration and thus could usefully dictate safer fishing sites for their popular trout derbies, for example. Unfortunately, when all salmonids were compared across the four geographic groupings, it was fish from island fishing sites closest to Gjoa Haven that had significantly higher Hg contamination than mainland sites (group 4), and ciscoes caught on the island were also significantly higher than mainland sites (Figure 3A). Island fishing sites are convenient, especially since access does not require travel over the Arctic ocean with its seasonal rough seas or sea ice. It is not known why island-sourced fish showed generally higher levels of Hg, but thawing permafrost and shallow lakes on the island could make fish that consume freshwater high trophic level prey especially vulnerable to the legacy of atmospheric Hg emissions.

As levels in the sampled salmonids varied over a very large range, more than two orders of magnitude (<0.5–270 mg·kg^−1^ dw), with this much variation seen in older lake trout alone (Figure 2B). Indeed, As concentrations were highest in lake trout, followed by Arctic char and cisco, with the lowest levels found in lake whitefish. As can bioaccumulate in fish [61], and as top predators with high growth rates, it is not surprising that lake trout and Arctic char showed the highest concentration of this contaminant. As concentrations in 64% of the salmonids exceeded the maximum level of 3.5 parts per million (ppm; mg·kg^−1^) in Canada’s Food and Drug Regulations [62]. However, the guideline applies to defatted fish protein from specific fish families in the orders Clupeiformes and Gadiformes (e.g., smelt and cod), and all the anadromous Arctic species analyzed here belong to the order Salmoniformes. Nonetheless, these As concentrations were higher than in other reports of the same species, including Arctic char at 0.017–13 µg·g^−1^ dw [60,63,64], lake whitefish at 0.07–2.8 µg·g^−1^ dw [65,66,67,68], and lake trout at 0.2–2.65 µg·g^−1^ dw [65,67,69]. Therefore, mean levels in older lake trout of 59 mg·kg^−1^ may be troubling and warrant further investigation. Overall, the As concentrations in these fish were higher than for other locations in the Arctic, with the highest concentrations similar to those seen in strictly marine fish [70]. Ocean fish have higher As levels, and much of this is in the form of arsenobetaine, which is considered non-toxic because it is not metabolized by humans [71,72]. Because the adult Arctic char fast in freshwater and consume only marine prey, it is likely that some of the As would have been in this form, at least in this species. A preliminary As species analysis with four Arctic char sampled from mainland sites showed that arsenobetaine accounted for 67–97% of the As present. Therefore, it would be prudent to conduct a more comprehensive speciation analysis, especially in lake trout, to ensure food safety.

When all the samples were grouped geographically, mainland fish (groups 3 and 4) had ~2-fold the As concentrations seen for the island group 2 and ~10-fold the concentration found for island group 1. Additionally, higher As concentrations in group 3 and 4 fish were seen in all individual fish species, and mean As concentrations of group 1 and 2 fish were consistently indistinguishable for all species (Figure 3). The higher concentrations of As in mainland fish could be explained by the mostly marine or estuarine fishing locations; as mentioned, fish from such locations have higher levels of arsenic, principally arsenobetaine.

As indicated, economic considerations limited the amount of PCB testing, but when the PCB data from all species were amalgamated, total PCB values varied from 5–367 µg·kg^−1^ ww, similar to the range of PCB concentrations in fish from other Arctic locations, including former military sites at 0.5–364 µg·kg^−1^ ww [73,74,75,76,77,78,79]. Lake trout and Arctic char showed the highest mean levels of PCBs, which can be explained by this contaminant’s known bioaccumulation and dependency on diet [80]. In this dataset, there was no correlation between fish PCB concentration and lipid content, contrasting with previous assumptions that fattier fish contain higher PCBs (e.g., [81], but also see [74]). The general observation that levels of the tested contaminants were higher in older fish was curiously reversed in lake whitefish, which live at the northern extreme of their distribution [50]. That these lake whitefish may be under stress is supported by otolith age analysis that shows that there are years where no juveniles were recruited. As well, the average condition (fish length/weight) of lake whitefish on the seasonal migration from ocean to lakes was significantly lower than the condition values at other times (not shown). Thus, we speculate that those lake whitefish with the additional burden of contaminant accumulation, including PCBs and Hg, may not live long lives, resulting in an apparent “decrease” in average PCB concentration in older fish. Indeed, sublethal levels of these chemicals have been observed to have a negative impact on another salmonid, rainbow trout, where a significant decrease in swimming performance and exercise recovery was seen following an injection of 100 µg·kg^−1^ PCBs, in the form of congener 126 [82].

Of the total salmonids analyzed for PCBs, 12% (12/101) exceeded freshwater sportfish guidelines (see Appendix A). However, guidelines for freshwater fish may not be directly applicable to anadromous fish. The total PCB concentrations are below the value (2000 µg·kg^−1^ ww) used by the U.S. FDA [83], but these guidelines are still under national review in Canada [55]. None of the 101 fish assayed contained dioxin-like (DL)-PCBs at concentrations that gave 2,3,7,8-tetrachlorodibenzo-p-dioxin (TCDD)-toxicity equivalency values exceeding those guidelines [62]. There was also no definitive geographical pattern, with the highest PCB concentrations in fish from all groupings and in all fish species (Figure 3C). Although there appears to be no cause for concern, it would be prudent to continue monitoring fish from the island sites in group 2 and the mainland group 3 sites, as they are the closest to former distant early warning stations (DEW Line) where PCBs were used, including Gladman and Matheson Points on King William Island as well as Shepherd Bay and Simpson Lake on the mainland (Appendix A). A similar recommendation has been noted for former Alaskan defense sites [79]. Although the DEW Line stations were remediated up to 5 decades after closure, PCBs may have contaminated the waterways in the interim.

The numbers and the positions of chlorines on the biphenyl structure correspond to different PCB congeners, and these can be used as “fingerprints” for PCB mixtures. Nonetheless, PCB congeners 110, 153, 118, and 138 had the highest abundance in all the fish, and three or more were amongst the five predominant congeners in 84% of all samples (Appendix A). Strikingly, congeners 153, 118, and 138 predominate in Monsanto Industrial Chemical Company’s Aroclor 1260, first manufactured in the 1950s, and these are routinely used as diagnostic congeners to identify this PCB mixture in soil samples (e.g., [57]). All three of these congeners also are found in Monsanto’s Aroclor 1254, along with PCB 110. These findings suggest that the sampled fish have retained the characteristic peaks of these two Aroclor mixtures, which were used at the DEW Line stations ([84]; Appendix A). The wide variation in PCB levels in different fish of the same species, even in the same geographic groupings, may very well reflect different levels of contamination in natal lakes in the region, another reason for further monitoring.

### 4.2. Genetic Tools for Geographical or Population Assignments and the Future of Contaminant Monitoring

Geographic groupings of contaminants showed variation, but there was a generally higher level of Hg in salmonids as a whole and ciscoes in particular from island fishing sites than on the mainland, with the reverse for As contamination in all sampled fish, as well as a clustering of distinct PCB congeners from lake whitefish and Arctic char from island sites (Figure 2 and Figure 3, and Appendix A). Theoretically, for routine contaminant analysis, it should be more cost effective and efficient for community members to net these anadromous species in the ocean during the salmonids’ summer feeding period. Samples could be taken for DNA, and diagnostic “kits” could quickly identify those fish originating from particular regions, such as island waters, which could then be targeted for specific contaminant testing. There would be no need for expeditions to dispersed fishing sites in other seasons with more challenging weather conditions, and the accrued savings could be used to assay more fish. To our knowledge, this strategy has not been previously undertaken, although the utility of as few as 20–150 SNPs for population assignment in mixed populations of fish in the ocean has been demonstrated [85,86]. We tested this idea by taking 413 Arctic char DNA samples at random to determine if they could be correctly assigned to each of 17 fishing sites. The results were disappointing, with overall few correct assignments to individual sites. This remained true irrespective of whether thousands of SNPs or only a subset of SNPs with the highest computed F_ST_ values were used. In the latter case, low assignment success characterized 65% of the fishing sites, with four sites approaching a surprising 0% success (Figure 5). This was likely attributable to the low genetic differentiation in Arctic char among sampled fishing sites, exacerbated by the small sample sizes at some locations.

In contrast, population assignments were much more successful with random samples from 413 Arctic char correctly assigned as “Northern” residents, those fish caught on or near King William Island, as well as a mainland or “Southern” population. An individual Arctic char could be correctly assigned to one of these two populations more than 85–90% of the time with only 4–32 SNPs, respectively (Figure 4). Therefore, rapid genetic assignments for Arctic char caught in the open ocean are feasible and suggest that new investigations and ongoing monitoring of contaminant levels using SNP markers could help target fish of possible concern, including, for example, those that could have higher As levels and some PCB congeners. Large panels of SNP markers are available for lake whitefish in this region with outlier analysis showing a few of these with potential to be linked to their natal sites [87]. If confirmed, such SNPs would be most helpful diagnostic tools for the higher PCB levels in this species from some island fishing sites. As yet, there has been no attempt to characterize populations of ciscoes or lake trout in this region, but it is not advised for this purpose alone. Ciscoes are not a favorite food fish, and considering that lake trout showed bioaccumulation of Hg, As, and PCB congeners, we reiterate our recommendation that lake trout consumption guidelines be developed by regional or territorial governments.

Our original hypothesis was that levels of contaminants would vary geographically and that genomic tools would be useful to help target food fish at risk, even in mixed stocks in the open ocean, circumventing the necessity to analyze hundreds of fish at particular fishing sites. This exploratory evaluation shows that analysis with as few as 4–32 genetic markers can correctly assign random Arctic char to geographically distinct regional populations. With additional testing, SNPs for lake whitefish may also hold promise. It is our hope that this novel strategy can be pursued in the future to allow more widespread testing of contaminants from these remote Arctic regions so as to help safeguard the wellbeing of Inuit who depend on these anadromous salmonids. Further, we expect that this approach will be applicable for the monitoring of vulnerable food fish elsewhere particularly in extreme locations.

## Figures and Tables

**Figure 1 foods-09-01824-f001:**
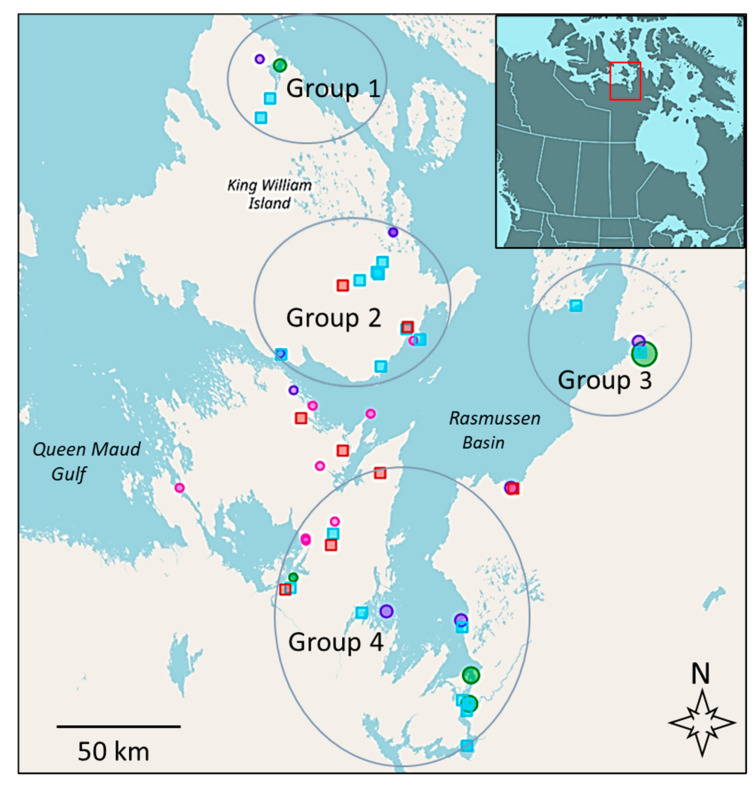
Map of the lower Northwest Passage region of the Kitikmeot in Nunavut (insert shows its location relative to a portion of North America), showing sites where fish were sampled and places of community concern for fish contaminants in subsistence food. Fish sampling sites are indicated by turquoise squares, clustered into four separate groups for statistical analysis: Group 1 for sites on the north of King William Island (sites 1 and 26), Group 2 for sites on the rest of the island close to the hamlet of Gjoa Haven at the southeast side of King William Island (sites 2, 3, 10, 17, and 21–24), Group 3 for fishing sites to the east of Rasmussen Basin (sites 8 and 14), and Group 4 for those on the southern mainland (sites 6, 7, 13, 18–20, 28–29). Additional non-traditional fishing sites used to enrich the genetic analysis, including site 31 are not shown. The map also shows open commercial fish harvesting sites (green circles, with the size of the symbol representing the permitted harvest), test fishing sites (pink or purple dots for currently open or closed test sites, respectively), and additional sites that the community would like to test for contaminants in the future (orange squares). Maps such as these are available as on-line atlases to the community and the public (https://tsfn.gcrc.carleton.ca/index.html?module=module.tsfnatlas.quotas), with some additional atlases are not yet publicly available due to community privacy concerns.

**Figure 2 foods-09-01824-f002:**
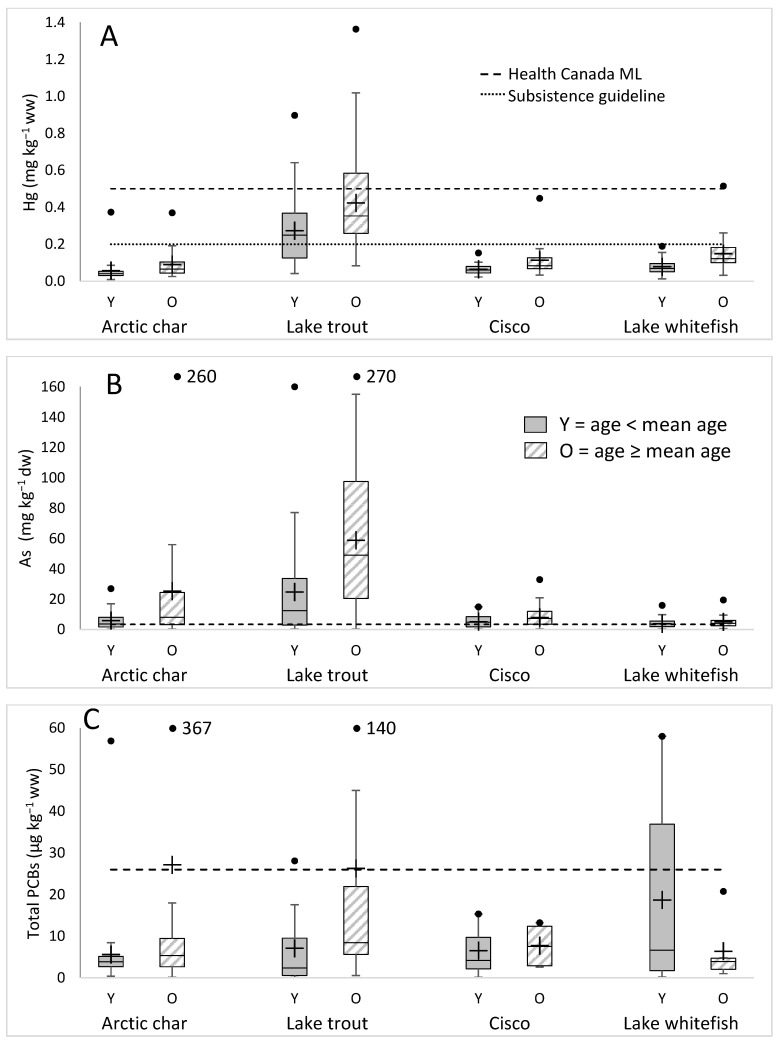
Box plots of contaminants assayed in different fish and grouped as either young (Y) or older (O) fish, defined as those younger than the mean age or at the mean age or older, respectively. (**A**). Hg (µg∙kg^−1^ ww), (**B**). As (mg∙kg^−1^ dw), and (**C**). total polychlorinated biphenyls (PCB) (µg∙kg^−1^ ww) concentrations with black points representing maximum values and crosses indicating means. Off-scale maximum values are provided in text boxes. Dotted horizontal lines are guidelines or advisories. See the Discussion for Health Canada maximum level (ML) or regulatory unit guidelines and other advisory information.

**Figure 3 foods-09-01824-f003:**
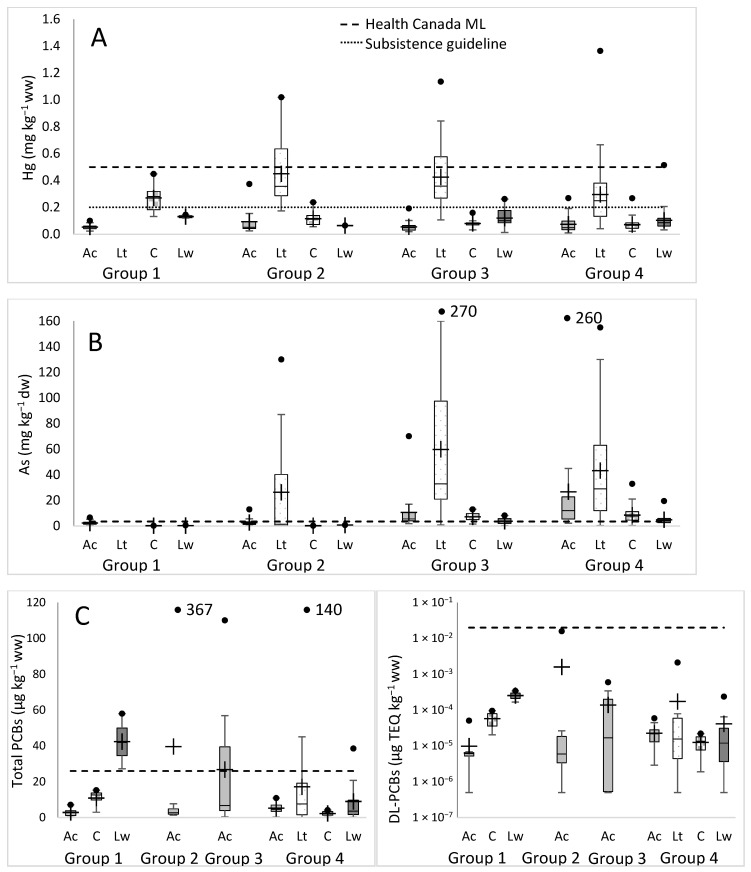
Box plots showing contaminants assayed in different fish and grouped geographically (as shown in Figure 1). (**A**). Hg (µg∙kg^−1^ ww), (**B**). As (mg∙kg^−1^ dw), and (**C**). PCB total concentrations (µg∙kg^−1^ ww) on the left, and DL-PCB (µg TEQ kg^−1^ ww) on the right. Fish are abbreviated as Ac = Arctic char, Lt = lake trout, C = cisco, and Lw = lake whitefish. Black points are maximum values and crosses are means. Off-scale maximum values are provided in text boxes. Dotted horizontal lines are guidelines or advisories. See the Discussion for Health Canada maximum level (ML) or regulatory limit guidelines and other advisory information.

**Figure 4 foods-09-01824-f004:**
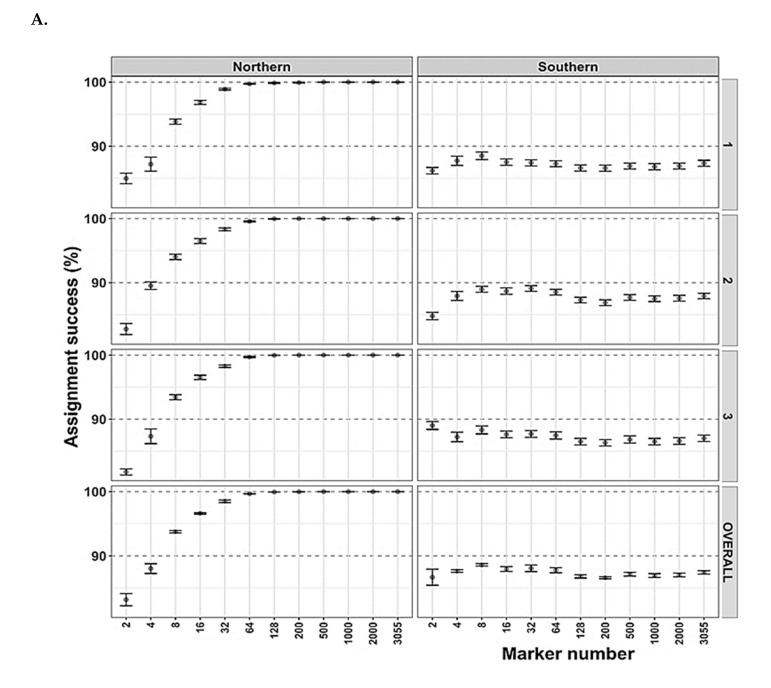
Different numbers (chosen as ~2 n, where *n* = 1 to 11) of single nucleotide polymorphisms (SNPs or DNA markers) as indicated by marker number were used to assign Arctic char samples to clusters representing either presumed populations or fishing locations. (**A**). Fish were assigned to “Northern” (left column) or “Southern” (right column) clusters, showing overall assignment success (y-axis) depending on the number of markers used and with the subsampling repeated three times as indicated by the successive stacked panels. Overall assignment success within the same cluster are shown on the bottom panel. (**B**). Arctic char were assigned to fishing site locations (indicated by site number according to Table 1, with fish at site 1 shown as spring- or winter-netted fish, 1_S or 1_W, respectively). Assignment success is shown relative to the number of markers used. An additional last panel shows the overall success of pooled results among sites. Similar figures analyzed with missing data imputed are found in Appendix A.

**Figure 5 foods-09-01824-f005:**
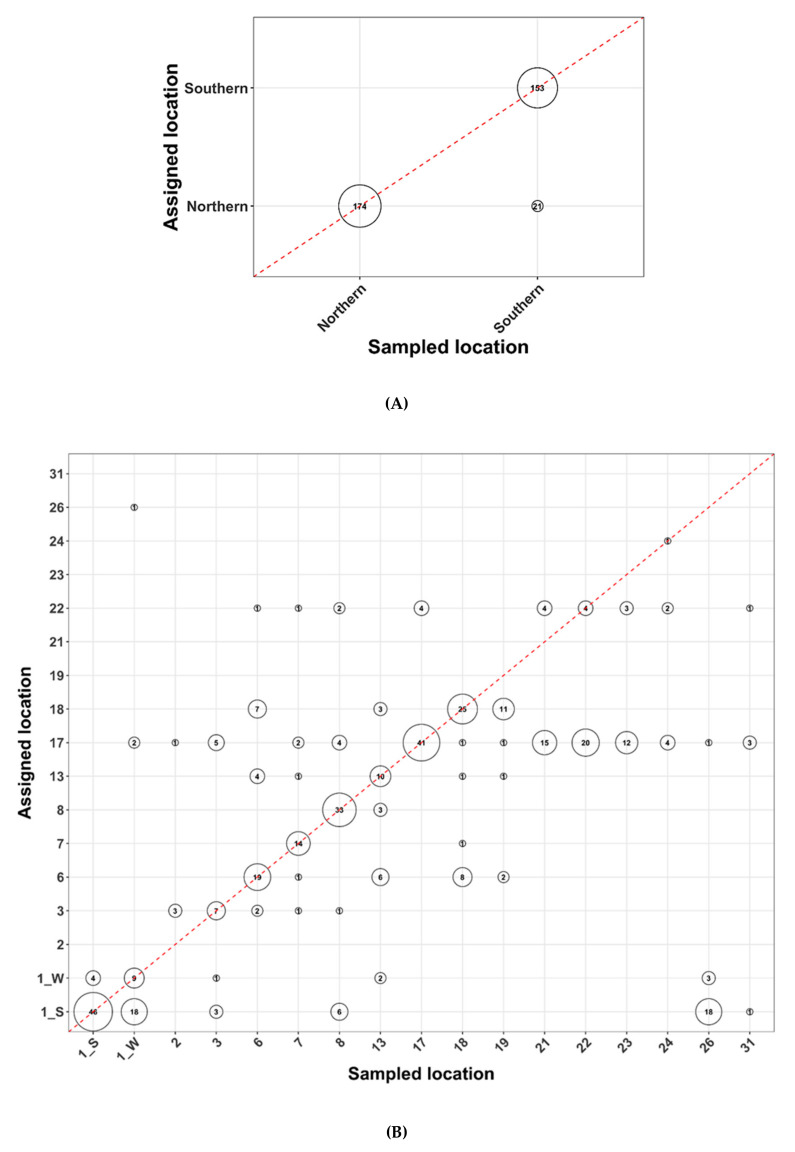
Assignment results with all Arctic char single nucleotide polymorphism markers (N = 3055). The circles on the red dashed diagonal represent successful assignments, and other circles display mis-assignments in which fish were not successfully assigned. Circle sizes are proportional to the number of fish. (**A**). Regional level assignment tests with Arctic char taken at random and designated either to “Northern” or “Southern” populations. (**B**). Fishing site assignment tests, with each individual fishing location indicated by a number, corresponding to the sites listed in Table 1 (with spring- and winter-netted Arctic char shown as 1_S and 1_W, respectively). Similar figures analyzed with any missing data imputed are found in Appendix A.

**Table 1 foods-09-01824-t001:** Fishing sites sampled in the Kitikmeot region of Nunavut, Canada, showing fishing site numbers and names or designations, global positioning system coordinates, geographic group assignments, as well as water type (fresh or saline), fishing gear used, and if samples from this location were analyzed for contaminants (**C**) and/or genetics (**G**).

Site Name & Number ^A^	GPS Location	Geographic Group	Water Type, Fishing Gear, Contaminants &/Or Genetics
1. Port Parry	69.55799, −97.43719	1	fresh/nets/C and G
26. GKWI lake 4	69.48518, −97.5351	1	fresh/nets/G
21. GKWI lake 1	68.85708, −96.46545	2	fresh/nets/G
22. GKWI lake 2	68.88999, −96.27868	2	fresh/nets/G
23. GKWI lake 3	68.88233, −96.25877	2	fresh/nets/G
17. KWI Weir	68.93280, −96.2195	2	fresh/spears/nets/C and G
2. Swan Lake	68.67045, −95.94928	2	fresh/nets/G
3. Koka Lake	68.53475, −96.21275	2	fresh/nets/C and G
10. Merilik Lake	68.57388, −97.32687	2	fresh/nets/C
24. Gjoa Haven	68.63374, −95.80885	2	saline/nets/G
8/14. Murchison R.	68.56700, −93.37707	3	fresh/nets/C and G
6. Backhouse Pt.	67.45756, −95.36072	4	saline/nets/C and G
7. Legendary R.	67.52161, −96.43939	4	saline/nets/C and G
13/18/19. Back R.	66.95853, −95.30144	4	fresh/nets/C
20. Hayes R.	67.13852, −95.2964	4	fresh/nets/G
28. W. of Chantry 3	67.81159, −97.04374	4	fresh/nets/C
29. W. of Chantry 4	67.86958, −96.7187	4	fresh/nets/C
31. Cambridge Bay ^B^	69.40600, −06.31393	-	fresh/nets/G

^A^ In some cases, sites were grouped when they were adjacent to one another (River is abbreviated to R., Point to Pt., King William Island to KWI, and GKWI lakes have not been assigned English names but were sampled for genetics). ^B^ For the purposes of this table and for the experiments described, Cambridge Bay fish for genetic analysis were arbitrarily assigned to site 31 and were not placed in a geographic group.

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
