# Peer review of "Identification of Arctic Food Fish Species for Anthropogenic Contaminant Testing Using Geography and Genetics"

_foods, 2020, doi:10.3390/foods9121824_

Round 1

Reviewer 1 Report

This is a fascinating read and for this referee, a new way of looking at these issues

The work is done carefully and the results are not in question, in my opinion.

Author Response

We thank the referee for the review of our manuscript, and we report that the paper has been further refined in this revision stage to achieve greater clarity for readers.

Reviewer 2 Report

This manuscript presents some interesting results about the levels of contaminants present in the edible tissue of commercial fish species in Canada.

I enjoyed reading most parts of the manuscript, the introduction supports the experimental work and the methodology is adequately described.

I have some suggestions which in my opinion should be considered by the authors.

The X-axis label of Fig 4 is upside-down. Moreover, the legend does not serve the reader, we need to present the content and help the reader see what we see here. Reference to the methodology can be brief and direct the reader to the relevant section for more details.

Some sections of the discussion are difficult to follow. Perhaps, an introductory sentence would help the reader. For example, section 3.3 starts with an explanation of a limitation of the present work. I prefer an introductory statement to present what will be justified in this paragraph. Start with what you can support and then describe the possible limitations. Most of the discussion needs to be revised to follow my suggestion if the authors agree.

The last part of the abstract and the relevant section of the discussion could be improved by doing a simple calculation on public health issue of Inuit people according the some population statistics on the average consumption of fish.  This will enable the reader and the scientific community to asses the risk of exposure to aquatic pollutants for Inuit people

Author Response

We thank the reviewer for the helpful suggestions.

1. The genomic figures have been revised according to the suggestions. Specifically, figures 4 and S7 have been changed to make them more readable by changing the panel arrangement format. 

2. Introductory lines: We have now revised the manuscript both in the Results and Discussion section to assist the reader. Specifically, paragraphs in sections 3.1., 3.2, 3.3, 3.4 and the Discussion paragraphs have been revised. If limitations were mentioned, they have been moved to the end of the paragraph where they will be better appreciated. This was a good suggestion; thank you.

3. Abstract and Discussion: This is an excellent point and we are thankful for the reviewer's suggestion.

(a) We have now revised the abstract to indicate that fish is a staple in the diet but unfortunately we were restricted in going into more detail since the length of that section is limited.

(b) We have fully addressed the question at the start of the Discussion. We now report that the residents harvested and consumed annually 9279 Arctic char, 2427 lake trout and 4080 whitefish and cisco according to the latest comprehensive wildlife harvest study (Priest and Usher 2004). This corresponds to approximately 22,500 kg of fish. Furthermore, using references for the estimated edible meat of these fish (Weihs and Okalik, 1989; Usher, 2000), and given an average population of 920 (Statistics Canada), this can then be calculated to provide about 108 servings of fish (of about one quarter of a kg per capita), which would represent more than two servings of locally harvested fish for every person per week. We have also added the appropriate references to the manuscript.  Again, thank you for the suggestion.

Reviewer 3 Report

Foods 966910

The manuscript entitled “Identification of arctic food fish species for anthropogenic contaminant testing using geography and genetics” has been submitted by Virginia K. Walker, Pranab Das, Peiwen Li, Stephen C. Lougheed, Kristy Moniz, Stephan Schott, James Qitsualik, Iris Koch.

In this study, the authors combined indigenous traditional knowledge and geography, assessment anthropogenic contaminant concentrations and fish genomic analyses. The idea is to target limited local resources after analysis of the presence of contaminants in order to make better use of them. To this end, the sampling of the study is carried out based on local knowledge of fishing locations and methods. Briefly, the authors have conducted their analyses on 4 fish species, Arctic char, lake trout, lake whitefish and ciscoes and three pollutants mercury, arsenic and polychlorinated biphenyls. Genomic analysis was undertaken for Arctic char only.

The study is scientifically sound, well conducted and meticulous. The study is of definite strategic interest by bringing together geographic and genetic data. The study is of strategic interest by bringing together geographic and genetic data to reduce the size of studies undertaken to measure the quality of foods potentially sullied by contaminants. Statistically, the study is robust.  The limits of the analysis are considered and analyses are carried out to adapt the appropriate statistical methods. The presentation of the strategy is interesting, but the strategy is disappointing for attributing the origins of the samples. The limits of this approach depend on the genetic diversity of the species studied. When applied to another species in anoter context, this strategy could be more effective. However, on a less precise scale, the method makes it possible to differentiate between the populations. Thus, in connection with the knowledge of the presence of contaminants, this study makes it possible to evaluate the quality of certain fishing sites. This study has the merit of presenting a methodological strategy, depending on the species studied (and their genetic diversity) and perhaps for other contexts, the objectives could be more ambitious. The study is well written, even if I am not qualified to evaluate this aspect of this work, but overall sentences a little long have sometimes slowed down the fluidity of reasoning.

  • Abstract : the objective of the study is not clearly stated.
  • Results (264-274) : average weight in kg or g ?

Author Response

We thank the reviewer for the comments and suggestions for improvement.

  1. As noted in the response to reviewer #2, we have now revised the manuscript so that the paragraphs in the Results and Discussion do not start with the limitations on the study but rather state the point of the paragraph with the other considerations placed latter. This should increase the fluidity of the text and make it easier for the reader. Thank you for this suggestion.
  2. We have revised the Abstract as suggested.
  3. We have revised the manuscript to correct the units in those lines (g); thank you!